# Prevalence of Malnutrition in Hospitalized Patients in Lebanon Using Nutrition Risk Screening (NRS-2002) and Global Leadership Initiative on Malnutrition (GLIM) Criteria and Its Association with Length of Stay

**DOI:** 10.3390/healthcare11050730

**Published:** 2023-03-02

**Authors:** Krystel Ouaijan, Nahla Hwalla, Ngianga-Bakwin Kandala, Emmanuel Kabengele Mpinga

**Affiliations:** 1Department of Clinical Nutrition, Saint George Hospital University Medical Center, Beirut 11002807, Lebanon; 2Institute of Global Health, University of Geneva, 1211 Geneva, Switzerland; 3Department of Nutrition and Food Sciences, American University of Beirut, Beirut 11072020, Lebanon; 4Department of Epidemiology and Biostatistics, Schulich School of Medicine and Dentistry, Western University, London, ON N6A 3K7, Canada; 5Division of Epidemiology and Biostatistics, School of Public Health, University of the Witwatersrand, Johannesburg 2000, South Africa

**Keywords:** malnutrition, nutrition assessment, nutrition screening, Global Leadership Initiative on Malnutrition (GLIM), Nutrition Risk Screening 2002 (NRS-2002), handgrip strength, mid-upper arm muscle circumference (MUAC), length of hospital stay (LOS)

## Abstract

(1) Background: Prevalence studies on hospital malnutrition are still scarce in the Middle East region despite recent global recognition of clinical malnutrition as a healthcare priority. The aim of this study is to measure the prevalence of malnutrition in adult hospitalized patients in Lebanon using the newly developed Global Leadership Initiative on Malnutrition tool (GLIM), and explore the association between malnutrition and the length of hospital stay (LOS) as a clinical outcome. (2) Methods: A representative cross-sectional sample of hospitalized patients was selected from a random sample of hospitals in the five districts in Lebanon. Malnutrition was screened and assessed using the Nutrition Risk Screening tool (NRS-2002) and GLIM criteria. Mid-upper arm muscle circumference (MUAC) and handgrip strength were used to measure and assess muscle mass. Length of stay was recorded upon discharge. (3) Results: A total of 343 adult patients were enrolled in this study. The prevalence of malnutrition risk according to NRS-2002 was 31.2%, and the prevalence of malnutrition according to the GLIM criteria was 35.6%. The most frequent malnutrition-associated criteria were weight loss and low food intake. Malnourished patients had a significantly longer LOS compared to patients with adequate nutritional status (11 days versus 4 days). Handgrip strength and MUAC measurements were negatively correlated with the length of hospital stay. (4) Conclusion and recommendations: the study documented the valid and practical use of GLIM for assessing the prevalence and magnitude of malnutrition in hospitalized patients in Lebanon, and highlighted the need for evidence-based interventions to address the underlying causes of malnutrition in Lebanese hospitals.

## 1. Introduction

Nutritional risk and malnutrition are highly prevalent in hospitalized patients [1], and have been reported to range from 20 to 50% in different European and South American countries with an average of 41.7% worldwide [2]. There is abundant evidence that malnutrition is associated with increased morbidity, nosocomial infections and hospital readmission [3]. Recent studies have also demonstrated that malnutrition is associated with prolonged length of stay (LOS) in patients with acute illness or even chronic non-communicable diseases [4,5]. Consequently, malnutrition is identified as a major encumbrance for hospitalized patients and a driver of increased healthcare cost incurring a considerable economic burden, accounting for 2.1 and 10% of the national health expenditures in Europe [6,7].

Nevertheless, malnutrition is still not addressed as a serious clinical problem due to the lack of clearly defined responsibilities and lack of unequivocally universally accepted diagnostic criteria [8,9]. Global efforts are being launched as well as a call to action to implement mandatory screening, establish a diagnostic code and develop national protocols to position nutrition as a healthcare priority [9,10]. Recently, the Global Leadership Initiative on Malnutrition (GLIM) has established a consensus for the diagnosis of malnutrition based on a combination of phenotypic and etiologic criteria and proposed it as a new tool to be validated in the disease-afflicted hospitalized population [11].

In the Middle East region, initiatives to study the prevalence of malnutrition in hospitals have been modest, with Turkey recently publishing a rate of 39% [12]. An international multicenter study published in 2008 has reported a lower rate of 22% of risk of malnutrition in two Lebanese hospitals [13]. Other prevalence studies in Lebanon have focused only on the rate of malnutrition in the community settings, with reported rates of 61.3% malnutrition and a risk of malnutrition in older adults living in long-term care centers and lower rates of 48.3% in older adults living in their homes [14,15].

### Context of the Study

Lebanon is a small country of the Middle East region covering an area of 10,452 km^2^ and having borders with both Syria and Israel, considered to be a conflict area. The country is divided into five main districts: north, Mount Lebanon, south, Bekaa Valley and the capital Beirut and its suburbs. In 2015, the population was estimated to be 6,847,712, including Lebanese people, foreign workers and refugees [16,17]. The highest population density is seen in Beirut and its suburbs. The south, north and Bekaa have the highest number of rural small villages.

Lebanon has one hundred and forty-four hospitals comprising 11742 beds, of which 78.3% are private and 21.7% are public. The number of beds is distributed as follows: 3806 (32.4%) in Mount Lebanon, 2452 in Beirut (20.9%), 1931 (16.4%) in the south, 1852 (15.8%) in the north and 1701 (14.5%) in Bekaa. The annual hospital admission is declared to be 698,210 cases per year, with the highest percentages in Beirut and Mount Lebanon, 22.3% and 29.6%, respectively [18]. 

According to the World Bank, the gross domestic product was estimated at USD 23.1 billion in 2021 compared to USD 52 billion in 2019. The drop in GDP per capita was a drastic 36.5% in just two years and Lebanon was reclassified as a lower-middle-income country instead of an upper-middle-income country. These drastic changes have resulted in difficulties in the cost of medical treatments and health coverage, which relies both on National Social Security and private insurances [17]. 

The aim of this study was to determine the prevalence of malnutrition in Lebanese hospitals by using the newly proposed GLIM tool, and to explore its different criteria and their relationship with length of stay, an easily measurable outcome parameter that is directly related to hospital costs [19]. The findings of this study will be the first milestone to establish a national policy mandating nutritional screening and assessment in all hospitalized patients. They can also guide the authority in forming a surveillance system and evaluating strategies targeted at decreasing the rate of malnutrition in hospitals.

## 2. Materials and Methods

### 2.1. Design and Sample Size

The study is a cross-sectional, observational, multicenter study. The sample size was estimated as 330 hospitalized patients to achieve a 95% confidence interval with a margin of error of 0.05 and 100% expected response rate based on using the STEPS sample size calculator of WHO and on the number of yearly hospital admissions [18]. It was calculated considering a significance level of 5% with 80% power. The number of patients in a random sample of hospitals in the five districts of Lebanon was weighed against the number of admissions per district from the National Health Survey [18]. The distribution of samples according to districts to have a national representation is presented in Figure 1. Private hospitals were only included due to the restricted access to the public hospitals in the period of data collection. All adult patients, males and females aged 18 years and above, admitted to the different wards of the hospital during the period of data collection were recruited within 48 h of admission. Exclusion criteria included the following wards: gynecology (including all pregnant and lactating women), intensive care unit, psychiatry and short stay of less than 48 h. 

### 2.2. Data Collection

Patient characteristics, i.e., age, gender, admission diagnosis, history of previous admissions, underlying diseases and number of home medications, were recorded. Patients were interviewed for history of weight loss, appetite and record of food intake. C-reactive protein levels (CRPs) were retrieved from the available blood tests from patients’ records. The length of hospital stay was calculated from the date of admission to the date of discharge. 

Body weight and height were measured using the Detecto manual scale to the nearest 1 kg and 1 cm, respectively. BMI (weight kg/height m^2^) was calculated accordingly. Mid-upper arm muscle circumference (MUAC) was measured at the midpoint between the acromion and olecranon processes at the non-dominant arm using a non-stretchable tape measure to the nearest 0.1 cm. The MUAC was categorized into three groups: “normal”, “moderately depleted” for measurements <23 cm and “severely depleted” for those <20 cm [20]. Handgrip strength was measured with the non-dominant hand using the Saehan hydraulic hand dynamometer to the nearest 0.1 kg. The handgrip strength variable was categorized into two groups: “normal” and “low” accounting for the gender cut-off points being <27 kg and <16 kg for males and females, respectively [20]. 

### 2.3. Nutritional Status

The Nutrition Risk Screening (NRS-2002) tool was used for nutritional screening, followed by an evaluation of malnutrition using the GLIM criteria. NRS is a two-step tool consisting of evaluating BMI, assessing recent weight loss and changes in food intake and identifying a grading of severity of disease as a reflection of increased nutritional requirements. Patients with a total score of 3 or more in the final screening were nutritionally at risk [21].

GLIM diagnosis was performed as a two-step process by firstly identifying at least one phenotypic criterion and one etiologic criterion and secondly assessing the severity of malnutrition as being either “moderate” or “severe” based on the phenotypic criterion [22]. Weight loss and BMI were used to evaluate the phenotypic criteria. The third phenotypic criterion evaluated was muscle mass, using MUAC as the measurement and handgrip strength as the supportive measure. MUAC was used as a surrogate technique as endorsed in recent recommendations in usual situations where body composition techniques such as bioelectrical impedance analysis and dual-energy X-ray absorptiometry are not available in the hospitals [23]. GLIM criteria emphasize that handgrip strength should be used as an additional supportive measure when only anthropometric measurements are available [22]. Handgrip strength is commonly employed in practice to assess muscle function qualitatively [23]. 

Reduced food intake, chronic gastrointestinal condition affecting absorption and inflammatory condition assessed via CRP levels were the etiologic criteria. Cut-off points of the different etiologic and phenotypic criteria are described in Table 1. 

### 2.4. Statistical Analysis

Statistical analysis was performed using STATA V17.1. Descriptive variables were described as n (%), mean ± standard deviation (SD) and median ± interquartile range (IQR). Cohen’s kappa (κ) was conducted to assess the agreement between NRS 2002 and GLIM. The length of hospital stay variable was then dichotomized into two groups with the median of 5 days used as the cut-off point: group one: ≤5 days and group two: >5 days. Mann–Whitney U and χ2 tests were performed to assess the differences in the length of hospital stay and history of hospital readmissions between the malnourished patients and those of normal nutritional status. Spearman’s rank correlations coefficient (rho) was used to measure the association between the non-parametric variables of length of hospital stay, handgrip strength and MUAC. Multiple logistic regression analysis was used to determine whether malnutrition with the GLIM criteria was independently associated with length of stay with adjustments for gender and admission diagnosis. All reported *p*-values were to a significance level of 5%.

### 2.5. Ethics

The study was completed in compliance with the guidelines of the Helsinki Declaration. The study protocol was reviewed and approved by the Institutional Review Board of the American University of Beirut (SBS-2020-0079). All participants reviewed and signed an informed consent form before participation. 

## 3. Results

### 3.1. Basic Characteristic

A total of 343 participants were enrolled in this study from May to October 2021. Baseline characteristics and distribution among districts are presented in Table 2. The mean age was 60 years (SD: 17 years) and the majority of the participants were less than 70 years old (65.89%). Surgical procedures (32.94%) and infectious diseases (27.7%) were the main diagnostic criteria for hospital admissions. 

### 3.2. Prevalence of Malnutrition

According to the NRS-2002 screening tool (Table 3), 31.20% of the participants had scores that were greater than or equal to 3 and thus were identified as being “at risk of malnutrition”, of which 51% were males and 49% were females. Beirut (38.27%) followed by the north (38.00%) and Mount Lebanon (33.00%) were the main districts identified by NRS-2002 as having participants at risk. The south had the lowest proportion (18.97%) compared to Beirut and the result was statistically significant (*p* = 0.016).

As for GLIM, 21.28% and 14.29% were identified as being “moderately” and “severely” malnourished, respectively, accounting for a total of 35.57% malnourished participants (Table 3). Half of the malnourished patients were male and the same proportion was female. Similarly to the NRS-2002 results identifying patients at risk of malnutrition, Beirut (43.21%), the north (42.00%) and Mount Lebanon (34.00%) were the main districts with malnourished participants (Figure 2). Bekaa had the lowest proportion (25.93%) compared to Beirut and the result was statistically significant (*p* = 0.043). 

The strength of the agreement between NRS 2002 and GLIM in identifying at-risk-of-malnutrition and malnourished patients as per Cohen’s kappa κ was 0.7580 (*p* < 0.001), indicative of good agreement.

### 3.3. Frequency of the Different GLIM Criteria

The frequencies of the different GLIM criteria among malnourished patients are described in Figure 3. Among the 122 patients who were identified as “moderately” and “severely” malnourished according to GLIM, the most dominant phenotypic criterion was “weight loss”, accounting for 82%. The median weight loss percentage was 8.5 kg (IQR 6.25–10). As for the etiologic criterion, the most prominent was “reduced food intake” accounting for 88% of patients, among which reduction in food intake for a period exceeding 2 weeks was the main measure (41.8%). The number of patients with low handgrip strength was 92 (75.4%). The mean handgrip strength of the males was 19.59 kg (SD = 4.28), whereas that of the females was 12.61 kg (SD= 2.44). As for the MUAC, 32 patients were identified as being moderately depleted (26.2%) and 10 patients were identified as being severely depleted (8.2%), a total of 42 patients (34.4%). The mean MUAC was 21.56 cm (SD = 0.7) and 20.2 (SD = 2.8) for males and females, respectively. More than half of the moderately malnourished patients had normal BMIs (54.9%).

### 3.4. Association of Malnutrition, Muscle Mass and Length of Hospital Stay

The patients’ median length of hospital stay was 5 days (IQR 3–10). There was a significant difference in the length of hospital stay between patients identified as malnourished according to GLIM criteria and those of normal nutritional status (11 days with IQR 9–15 versus 4 days with IQR 3–5, respectively, *p* < 0.001). When a median of 5 days was considered as the cut-off point, 90.9% of malnourished patients had a length of hospital stay greater than 5 days compared to 9.1% of patients of normal nutritional status, as shown in Table 4 (*p* < 0.001).

Handgrip strength and MUAC measurements were negatively correlated with the length of hospital stay (rho/ρ = −0.40, *p* < 0.001 and rho/ρ = −0.25, *p* < 0.001), regardless of the patient’s nutritional status. Patients with low handgrip strength measurements had a length of hospital stay greater than the median of 5 days (74.4% versus 25.6%, *p* < 0.001). As for patients with moderate and severe depletion in MUAC measurements, 84.4% had a length of hospital stay greater than the median (84.4% versus 15.6%, *p* < 0.001) (Table 4).

### 3.5. Multiple Logistic Regression of Length of Hospital Stay

Having a malnutrition diagnosis was found to be an independent predictor of length of hospital stay, as shown in Table 5. Specifically, patients who were identified as malnourished according to GLIM criteria (*p* < 0.001) had higher odds of having a length of hospital stay that exceeded 5 days compared to those who were well-nourished. Age was excluded from the model because it was part of the malnutrition diagnosis. The Hosmer and Lemeshow goodness-of-fit test indicated that our model fit the data well with *p*-values of 0.2364.

### 3.6. Association of Malnutrition with Hospital Readmission

Patients who were identified as being malnourished according to GLIM criteria (33.61%) were more likely to have been previously admitted to the hospital in the past 3 months compared to those identified as having a normal nutritional status (3.17%) (χ2 = 60.51, *p* < 0.001).

## 4. Discussion

The prevalence rate of malnutrition risk among hospitalized patients was 31.2% according to NRS-2002 and the prevalence of malnutrition according to the GLIM criteria was 35.6%. These figures is different from previous data collected in 2008 in two large Lebanese hospitals of the international multicenter study, where malnutrition risk was only screened and the rate was 22% using the NRS-2002 tool [13]. In addition to the fact that our data are larger and more hospitals were included, this difference in rate reflects the increase in the risk of malnutrition in hospitalized patients in a country where economic crisis has drastically deteriorated. This crisis is affecting the access to and availability of nutrition care in hospitals [17]. 

The higher percentage of malnutrition according to GLIM was detected in the capital Beirut (43.2%), where hospitals are larger and more complicated cases are admitted. A lower prevalence of 26% was observed, on the other hand, in Bekaa where the population density is much lower [18]. The prevalence in the five districts is very similar to the rates reported in other countries, varying from 20% to 50% with higher ranges in developing countries [2,12]. One other recent study restricted to one hospital in Lebanon with a smaller sample size reported that 34.7% of their sample population was at risk of malnutrition and 9.3% were malnourished [24]. Although the percentage of at-risk patients is high, their lower rate of malnutrition is probably due to the use of a different tool, which was the Mini Nutritional Assessment MNA, specific to older adults [24].

The prevalence of risk of malnutrition when using NRS-2002 was slightly lower than the prevalence rate of the malnutrition diagnosis using GLIM criteria, reporting a rate of 31.2%. However, there was a good agreement statistically between the two tools. This concordance was also recently reported in a study on hospitalized patients in Turkey, where GLIM was correlated with NRS-2002 and not with other nutrition assessment tools [25]. Other studies have found a stronger correlation between GLIM and other screening tools such as the Malnutrition Universal Screening Tool (MUST), but the sample population was of older adults and those specifically having cancer [26,27,28]. Therefore, NRS-2002 is still considered to be a valid and more specific tool to be used for hospitalized patients during the screening process as recommended by clinical practice guidelines [29].

GLIM is considered to be a diagnostic tool to be used after screening to confirm nutritional assessment. It is different from other assessment tools as it has many different criteria and severity levels. In our study, we have studied the frequency of each phenotypic and etiologic criterion in patients diagnosed with moderate and severe malnutrition. The most frequent criteria were weight loss and low food intake, which are quick and easy to collect. This same combination of weight loss and low food intake was observed in a study on the validation of GLIM and was considered to be the most predictive with regard to worse clinical outcomes [30]. On the other hand, low BMI in our sample population was the least recorded criterion, with 16% compared to 88% for weight loss and 57% for low muscle mass. More than half of malnourished patients had a normal BMI, reemphasizing the importance of not relying solely on BMI in nutrition assessment, an issue always challenged by clinicians [31]. 

Patients identified as malnourished by GLIM had a significantly longer length of stay (LOS) of 7 days and had significantly higher rates of previous hospital readmissions. Both LOS and the incidence of hospital readmissions are surrogate markers of a patient’s clinical outcomes and economical costs [32,33]. This strong correlation associates malnutrition with unexpected complications and a worsening clinical status of patients, highlighting the importance of identifying malnutrition early during hospitalization. The prediction model identifying malnutrition diagnosis as a predictor of length of stay independent of underlying diseases reinforced the association of malnutrition with worsening clinical outcomes. It demonstrates the validity of GLIM criteria to predict prolonged hospitalization as a health outcome [34]. 

Interestingly, a correlation with LOS was also found in our study with low MUAC and handgrip strength, independently of nutritional status. Handgrip strength has previously been linked to longer hospitalization but MUAC has never been studied from this perspective since it is commonly more used in the pediatric population [35,36]. Our findings may help in adding simple anthropometric measurements not requiring expensive tools such as MUAC in assessing muscle mass as part of GLIM criteria when body impedance analysis (BIA) or dual-energy X-ray absorptiometry DEXA are not available [37].

Our study findings of high prevalence rates support the need for increasing awareness towards malnutrition, which many global efforts are now targeting. Consequently, the newly developed European Nutrition for Health Alliance has started the Optimal Nutritional Care for All (ONCA) campaign, which launched a global call for action in 2013 to all countries to raise public awareness, establish a nutrition assessment pathway and develop national protocols to include effective nutrition care as a fundamental right to heath [16]. Other similar associations from different countries followed this path and launched an international call to action in a forum “Linking Nutrition Around the World” [9]. In addition, the United Nations Decade of Action on Nutrition emphasized that national policies should prioritize aligned health systems providing universal coverage of all essential nutrition actions [38]. Lebanon and other countries in the Middle East have not joined these global efforts yet. However, a national policy, supported by international instruments, is becoming a necessity to identify and target malnutrition, especially in the economic crisis that the country is going through. 

It is important to mention that initiatives and policies targeting malnutrition should recognize the crucial role of dietitians in the nutrition care of the patient [39]. Clinical dietitians are integral members of the multidisciplinary team in the hospitals and they are uniquely qualified in the assessment and the management of malnutrition in the care pathway of the patients [40,41]. They are specialized in interpreting anthropometric measurements, recommending nutrition support plans and providing informational counseling to patients [39,42]. Their nutrition interventions will aim to improve the continuum of care of the hospitalized patients in enhancing clinical outcomes.

### Strength and Limitations

To our knowledge, this is the first study to report the prevalence of malnutrition in hospitalized patients in a national representative sample of hospitals in Lebanon and is one of the very few studies in the Middle East. Nutrition screening and assessment were conducted upon admission in a heterogeneous population of different medical and surgical diagnoses, making our study different from other prevalence studies conducted retrospectively and on a specific patient population. The GLIM tool that is newly developed was also used with simple anthropometric measurements that could be easily found in settings with minimal resources. Our study nevertheless has limitations. Data were collected from private hospitals only and public hospitals were excluded due to security reasons, meaning that patients admitted to these hospitals of usually lower socioeconomic status were not represented. The cut-off values we used for MUAC and handgrip strength to assess muscle mass were taken from consensus recommendations and were not validated in different patient populations. We therefore recommend that future studies clarify their cut-off values.

## 5. Conclusions

Our present study reports a considerable high prevalence of malnutrition in hospitalized patients upon admission that was directly associated with a longer length of stay, implicating worsening clinical outcomes. Since the identification of malnutrition remains an important first step to target its recognition and management in daily clinical practice, the use of GLIM criteria with simple, affordable and anthropometric measurements is considered to be both valid and a practical diagnosis step.

## Figures and Tables

**Figure 1 healthcare-11-00730-f001:**
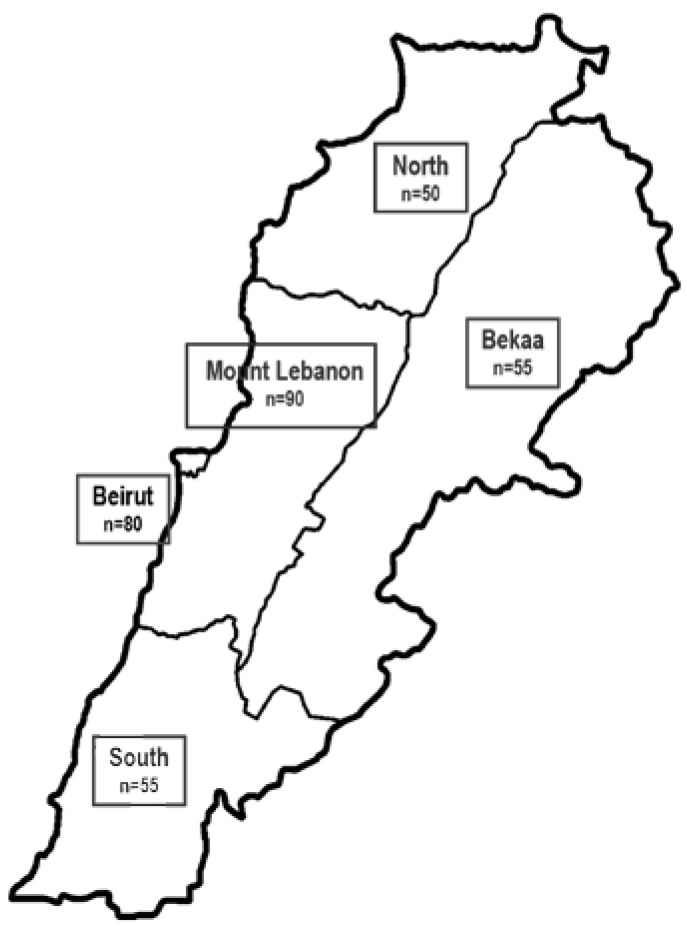
Distribution of sample according to district representation.

**Figure 2 healthcare-11-00730-f002:**
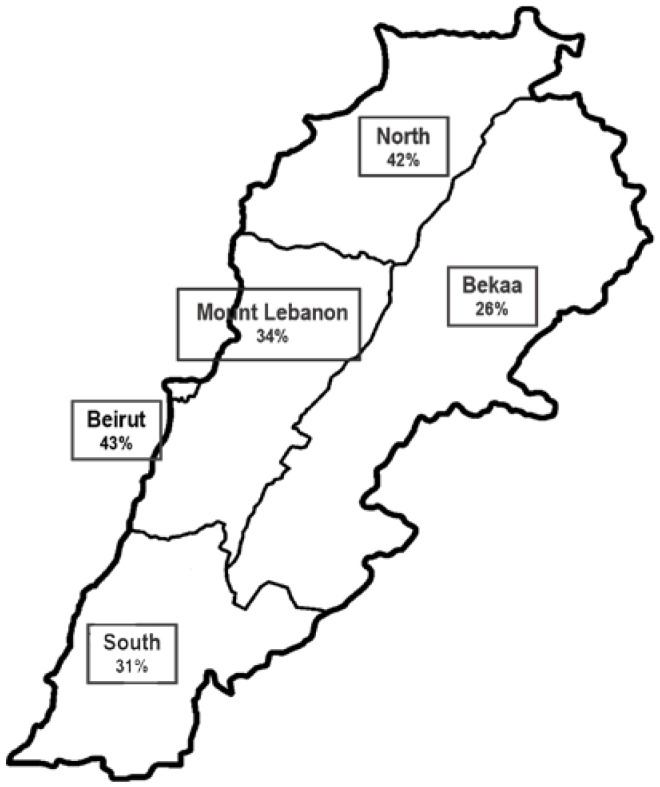
Distribution of malnutrition rates in the different districts according to Global Leadership Initiative on Malnutrition (GLIM).

**Figure 3 healthcare-11-00730-f003:**
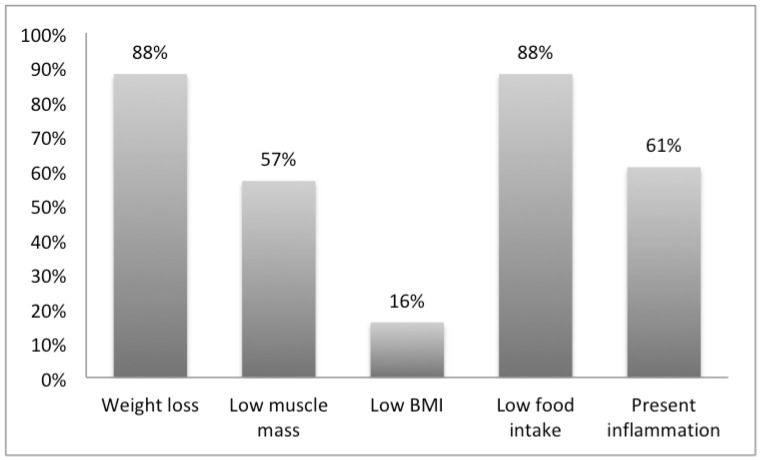
Frequencies of the phenotypic and etiologic criteria present in the malnourished patients as classified by the Global Leadership Initiative on Malnutrition (GLIM) (n = 122).

**Table 1 healthcare-11-00730-t001:** GLIM criteria for diagnosis of malnutrition [22].

Phenotypic Criteria			Etiologic Criteria	
Severity level	Moderate	Severe		
Weight loss	>5–10% within past 6 months or 10–20% beyond 6 months	>10% within past 6 months or >20% beyond 6 months	Reduced food intake	<50% of estimated needs in >1 week or any reduction for >2 weeks
Low BMI	<20 kg/m^2^ if <70 years, <22 kg/m^2^ if ≥70 years	<18.5 if <70 years, <20 if <70 years		Any chronic GI condition that adversely impacts food assimilation or absorption
Reduced muscle mass ^2^	MUAC ^1^ < 23 cm	MUAC < 20 cm	Inflammation	Elevated C-reactive protein (CRP) levels

^1^ Mid-upper arm muscle circumference (MUAC). ^2^ Handgrip strength was used as the supportive measure as recommended.

**Table 2 healthcare-11-00730-t002:** Baseline characteristics of participants (N = 343).

Characteristic	N (%)
Age	
<70 years old	226 (65.9%)
≥70 years old	117 (34.11%)
Gender	
Male	188 (54.81%)
Female	155 (45.19%)
District	
Beirut	81 (23.62%)
North	50 (14.58%)
South	58 (16.91%)
Mount Lebanon	100 (29.15%)
Bekaa Valley	54 (15.74%)
Present illness	
Oncology	25 (7.29%)
Cardiovascular disease	53 (15.45%)
Infectious disease	95 (27.70%)
Gastrointestinal disease	40 (11.66%)
Surgical procedure	113 (32.4%)
Other	17 (4.96%)
Underlying disease	
None	87 (25.36%)
Diabetes and cardiovascular diseases	192 (55.98%)
Cancer	44 (12.83%)
Neurological disorders	8 (2.33%)
Gastrointestinal diseases	12 (3.50%)
Home medications	
None	110 (32.07%)
One medication	46 (13.41%)
Two medications	46 (13.41%)
Three or more medications	141 (41.11%)
Previous hospital admission within 3 months	
Yes	48 (13.99%)
No	295 (86.01%)

**Table 3 healthcare-11-00730-t003:** The prevalence of malnutrition according to Nutrition Risk Screening (NRS-2002) and Global Leadership Initiative on Malnutrition (GLIM) (N = 343).

Prevalence Rate	N (%)
NRS-2002	
Mild risk (<3)	236 (68.8%)
At risk (≥3)	107 (31.2%)
GLIM	
Normal nutritional status	221 (64.43%)
Malnourished	122 (35.57%)
Moderate malnutrition	73 (21.28%)
Severe malnutrition	49 (14.29%)

**Table 4 healthcare-11-00730-t004:** Effect of various measurements of nutritional status on length of stay (LOS) greater than 5 days in Lebanese hospitals (n = 343).

Nutritional Status	Low	Normal
Nutritional status according to GLIM criteria ^1^	90.9%	9.1%
Handgrip strength (HGS)	74.4%	25.6%
Mid-upper arm muscle circumference (MUAC)	84.4%	15.6%

^1^ Global Leadership Initiative on Malnutrition.

**Table 5 healthcare-11-00730-t005:** Multiple logistic regression models for length of stay.

	Odds Ratio (OR)	95% CI for OR	*p*-Value
Underlying disease ^a^			
Diabetes and cardiovascular diseases	0.97	0.31; 3.06	0.963
Cancer	1.50	0.31; 7.11	0.608
Other (neurological disorders and gastrointestinal diseases)	0.56	0.11; 2.87	0.489
Home medications ^b^			
1–2	1.67	0.56; 5.03	0.356
≥3	1.08	0.32; 3.67	0.894
Present illness ^c^			
Oncology	5.00	0.86; 29.05	0.073
Cardiovascular disease	4.85	0.92; 25.68	0.063
Infectious disease	0.87	0.13; 5.78	0.889
Gastrointestinal disease	1.16	0.22; 5.97	0.854
Other	3.18	0.39; 25.58	0.276
Malnutrition diagnosis ^d^			
Present	60.72	23.97; 153.78	<0.001 *

^a^ Reference group “none”, ^b^ reference group “none”, ^c^ reference group “none”, ^d^ reference group “absent”. * *p* < 0.05.

## Data Availability

Data are contained within the article.

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
