# Peer review of "Prevalence of Malnutrition in Hospitalized Patients in Lebanon Using Nutrition Risk Screening (NRS-2002) and Global Leadership Initiative on Malnutrition (GLIM) Criteria and Its Association with Length of Stay"

_healthcare, 2023, doi:10.3390/healthcare11050730_

Round 1
Reviewer 1 Report
Review of healthcare-2214527 Article Title: Prevalence of malnutrition in hospitalized patients in Lebanon using Global Leadership Initiative on Malnutrition GLIM criteria and its association with length of stay The authors undertook a an interesting topic about prevalence of malnutrition in hospitalized patients in Lebanon. The content of the manuscrpit, however, has few reservations. · The title mistakenly suggests that the suggests that the frequency of malnutrition was assessed only on the basis of the GLIM test, however, in the research methodology, the authors provide two equally valuable tools: GLIM and NRS 2002 tests - the title should be changed. · The Keywords section - „Length of Hospital Stay” has no abbreviation · In the Introduction section, there is no information on what percentage of the entire Lebanese population is malnourished and what this condition may result from. · In the Design and sample size section, there is no information about the percentage of patients from public and private hospitals (this information only appears in the Limitations section). Exclusion criteria are imprecise - were pregnant or in the last year post-pregnancy women included in the study, or hose whose anthropometric measurements could not be obtained due to the amputation of limbs (superior and/or inferior)? · It is puzzling why the authors did not confirm malnutrition with simple biochemical indicators such as protein, albumin or CLL concentrations. · Baseline characteristics of participants lack information about patients' place of residence (city or village) and income. This information could affect the cause of malnutrition - low sociodemographic status or resulting from an underlying disease. · In the Discussion section, it is worth including information on how the results of this study can help improve the well-being of patients, whether nutritional treatment will be implemented in them. The manuscript needs minor revision.
Author Response
Thank you for the valuable review and much important feedback, We have taken into account the comments in our revision. Kindly find below a detailed response.
Point 1: The title mistakenly suggests that the suggests that the frequency of malnutrition was assessed only on the basis of the GLIM test, however, in the research methodology, the authors provide two equally valuable tools: GLIM and NRS 2002 tests - the title should be changed.
Response 1: We have changed the title to include both GLIM and NRS.
Point 2: The Keywords section - „Length of Hospital Stay” has no abbreviation.
Response 2: We have added the abbreviation LOS.
Point 3: In the Introduction section, there is no information on what percentage of the entire Lebanese population is malnourished and what this condition may result from.
Response 3: Since there is no national data on rate of malnutrition in the adult population and data is only on children, we have added two recent published studies on the rate of malnutrition in the older adults in the community in the third paragraph in the introduction. These studies will add as suggested a better background on the nutritional status of the Lebanese population. In addition, we have highlighted the economic situation of Lebanon in the context of the study.
Point 4: In the Design and sample size section, there is no information about the percentage of patients from public and private hospitals (this information only appears in the Limitations section). Exclusion criteria are imprecise - were pregnant or in the last year post-pregnancy women included in the study, or hose whose anthropometric measurements could not be obtained due to the amputation of limbs (superior and/or inferior)? ·
Response 4: We have added now that only private hospitals were included due to access problems. As for exclusion, we have also clarified the criteria for the gynecology ward. We have also planned to take anthropometrics on all patients and adjust for amputations. However we had no such cases in our sample.
Point 5: It is puzzling why the authors did not confirm malnutrition with simple biochemical indicators such as protein, albumin or CLL concentrations.
Response 5: After many discussion between the research team, a decision was taken not to include albumin and other similar biochemical markers due to the GLIM recommendation of not having them as part of criteria. In addition new evidence and guidelines have removed these markers from the nutrition assessment due to their low sensitivity being negative phase proteins (Mendez et al. Nutr Clin Pract 2005 20: 314) (Evans et al. Nutr Clin Pract 2020; Oct. 1-7). The research group after consulting the GLIM working group relied on the criteria of GLIM that include CRP in their etiologic part.
Point 6: Baseline characteristics of participants lack information about patients' place of residence (city or village) and income. This information could affect the cause of malnutrition - low sociodemographic status or resulting from an underlying disease.
Response 6: Sociodemographic and other social determinants were collected as part of sub analysis that will be analyzed in a different context. The research group wanted to focus on the clinical aspect of malnutrition and its effect on LOS. The sub-analysis to discuss social determinants of hospital malnutrition will be done as suggested as next step.
Point 7: In the Discussion section, it is worth including information on how the results of this study can help improve the well-being of patients, whether nutritional treatment will be implemented in them.
Response 7: We have added a distinct paragraph at the end of discussion to raise this very important issue, the role of dietitian in providing different facets nutrition care in patient's stay and its effect on improving malnutrition.
We thank you again a lot for the review and the suggestions were crucial to be added to our revision. We remain at your disposal for any further edits or inquiries needed.
Reviewer 2 Report
The manuscript titled with " Prevalence of Malnutrition in Hospitalized Patients in Lebanon 2 using Global Leadership Initiative on Malnutrition GLIM cri- 3 teria and its Association with Length of Stay ". The manuscript discuss a good points. Overall, The present study reports a considerable high prevalence of malnutrition in hospitalized patients upon admission that was directly associated with longer length of stay suggesting worsening clinical outcomes. Identification of malnutrition by routine screening and thorough diagnosis remain an important first step to target its recognition and management in daily clinical practice. The use of GLIM criteria with simple and affordable and anthropometric measurements is considered both valid and practical. The manuscript written well. But it needs a minor revision. The materials and methods should be shortening and also the conclusion should be concentrated in two or three sentences.
Author Response
Thank you for the valuable review and feedback. We have taken the suggested edits into consideration in the revision. Kindly findly below a more detailed response.
Point 1: The materials and methods should be shortening and also the conclusion should be concentrated in two or three sentences.
Response 1: Due to the needed information concerning sample size and a detailed description of anthropometrics in GLIM required by review, we could not shorten the materials and methods section. As for the conclusion, we have reformulated into two concentrated sections.
Thank you again for the review and we remain at your disposal for any further suggestions.
Reviewer 3 Report
The work is very interesting ai valuable. However, I think it is worth making some minor improvements.
1. introduction
It is worth adding that previous studies have studied the factors associated with prolonged LOS for several common diseases e.g..
https://doi.org/10.1016/j.gaceta.2016.01.003
https://doi.org/10.3390/ijerph19105827
https://doi.org/10.3390/jcm10163715
The purpose of the study should be included at the end of the introduction.
2. Methodology
The Ethics section should be included at the end of the manuscript as required by the journal - see template
3. the results are presented correctly and clearly
4. the discussion is correct
5. the strengths and weaknesses of the study are described correctly and demonstrate the scientific maturity of the researchers.
6. conclusions are formulated correctly and supported by the results.
Author Response
Thank you so much for the valuable review and feedback. We have done edits and additions to the article accordingly. Kindly find below our response:
Point 1: It is worth adding that previous studies have studied the factors associated with prolonged LOS for several common diseases e.g.. The purpose of the study should be included at the end of the introduction.
Response 1: A statement on studies studying LOS and malnutrition in different diseases has been added in the first paragraph of the introduction. Two references suggested were added. In addition, the aim of the study was emphasized by as suggested being moved to end of introduction.
Point 2: The Ethics section should be included at the end of the manuscript as required by the journal.
Response 2: A section was there at the end of manuscript stating the IRB statement. It was now edited and changed to an Ethics statement.
Thank you again for the review and we remain at your disposal for any further edits and inquiries.
Reviewer 4 Report
The article presented for review is a research report. I have a few comments:
1. In the „Introduction”, authors should include the aim of the study.
2. In “Materials and Methods”:
a. BMI calculation should be included
b. Line 118 – units are missing
c. Table 1 - - units are missing. There is a mistake in the criteria “low BMI.”
d. Authors should precisely describe the GLIM criterium “reduced muscle mass”. It is crucial because other methods than MUAC are also possible (ASMI, FFMI, ALM, …).
e. In table 1. The criterium “reduced muscle mass” is described as MUAC. The authors must clearly explain the reason for using handgrip strength as assessment support. It is not clear.
3. In “Results”
a. Figure 3, Table 4 – The number of participants should be included.
The results have been described and appropriately presented. Conclusions have been prepared correctly.
Author Response
We thank you for the helpful review and valuable feedback.
Kindly find below our responses and edits we have done:
Point 1: In the „Introduction”, authors should include the aim of the study.
Response 1: The aim of study was mentioned not clearly from our part in one paragraph of the introduction. We have reformulated the aim and highlighted in one separate paragraph at the end of the introduction.
Point 2: Changes required in Materials and Methods
a. BMI calculation should be included
Response 2 a: BMI calculation was added in section 2.2 in Methods
b. Line 118 – units are missing
Response 2b: The units of handgrip strength that were missing were added in kg in section 2.2 in Methods.
c. Table 1 - - units are missing. There is a mistake in the criteria “low BMI.”
Response 2c: Units and criteria of MUAC and BMI were corrected and added in Table 1.
d. Authors should precisely describe the GLIM criterium “reduced muscle mass”. It is crucial because other methods than MUAC are also possible (ASMI, FFMI, ALM, …)
e. In table 1. The criterium “reduced muscle mass” is described as MUAC. The authors must clearly explain the reason for using handgrip strength as assessment support. It is not clear.
Response 2 d&e: It was much needed to add thanks to your review the explanation on our used of MUAC and Handgrip strength in our methods especially that it is one of our discussion points. Since DEXA and BIA were not available in our reserach and in the clinical settings of data collection to measure FFMI and ASMI, we have used the MUAC supported by handgrip strength as surrogate measures as recommended by the GLIM working group in their article. We have added in the Methods in Section 2.3 the missing explanation. We have mentioned that the standard methods were not available and that we have followed the GLIM recommendations. We have also explained that handgrip strength is used as supportive measure since MUAC is not considered a gold standard. We have added that new recommendation from Prado et al endorse the use of anthroprometric measurements such as MUAC and HGS. We have added the article recently published in Clinical Nutrition entitled: "Advances in muscle health and nutrition: A toolkit for healthcare professionals" in our references.
Point 3. Figure 3, Table 4 – The number of participants should be included.
Response 3: Number of participants was added as n in both Table 4 (all included participants) and Figure 3 (malnourished patients)
Thank you again for your review and we remain at your disposal for any further inquiries.